# The Adaptive Mechanism of Plants to Iron Deficiency via Iron Uptake, Transport, and Homeostasis

**DOI:** 10.3390/ijms20102424

**Published:** 2019-05-16

**Authors:** Xinxin Zhang, Di Zhang, Wei Sun, Tianzuo Wang

**Affiliations:** 1State Key Laboratory of Vegetation and Environmental Change, Institute of Botany, The Chinese Academy of Sciences, Beijing 100093, China; zhangxinxin19870620@hotmail.com (X.Z.); zhangdi@ibcas.ac.cn (D.Z.); 2College of Resources and Environment, University of Chinese Academy of Sciences, Beijing 100049, China; 3Key Laboratory of Vegetation Ecology, Ministry of Education, Institute of Grassland Science, Northeast Normal University, Changchun 130024, China; sunwei@nenu.edu.cn

**Keywords:** iron deficiency, acquisition, transport, homeostasis

## Abstract

Iron is an essential element for plant growth and development. While abundant in soil, the available Fe in soil is limited. In this regard, plants have evolved a series of mechanisms for efficient iron uptake, allowing plants to better adapt to iron deficient conditions. These mechanisms include iron acquisition from soil, iron transport from roots to shoots, and iron storage in cells. The mobilization of Fe in plants often occurs via chelating with phytosiderophores, citrate, nicotianamine, mugineic acid, or in the form of free iron ions. Recent work further elucidates that these genes’ response to iron deficiency are tightly controlled at transcriptional and posttranscriptional levels to maintain iron homeostasis. Moreover, increasing evidences shed light on certain factors that are identified to be interconnected and integrated to adjust iron deficiency. In this review, we highlight the molecular and physiological bases of iron acquisition from soil to plants and transport mechanisms for tolerating iron deficiency in dicotyledonous plants and rice.

## 1. Introduction

Iron (Fe) is an essential micronutrient for plant growth development and plays a key role in regulating numerous cellular processes. Iron, as an important co-factor for enzymes, plays an important role in regulating plant photosynthesis, mitochondrial respiration, the synthesis and repair of nucleotides, and metal homeostasis, especially in the maintenance of structural integrity of various proteins [1]. While Fe is abundant in soil, the available Fe in soil for plants is often insufficient, particularly in calcareous soils, due to low solubility of Fe. Iron deficiency is one of the most important factors limiting crop production in the world. Plants grown in low Fe soils often exhibit chlorosis and decreased photosynthesis, leading to reduction in yield and quality of crops. To cope with this situation, plants have evolved a series of sophisticate mechanisms to adapt to iron-deficient conditions in soil. In addition, iron deficiency is a significant worldwide problem, seriously affecting over 30% of the world’s population (http://www.who.int/nutrition/topics/ida/en/). Anemia as one of the severest nutritional disorders is caused by low iron in humans. Therefore, elucidation of the molecular and physiological mechanisms by which plants sense, respond, and adapt to Fe deficiency would contribute to cultivating crop varieties with high Fe efficiency.

## 2. Iron Acquisition from Soil to Roots

Although iron is considered as the fourth most abundant element, one-third of soil on the Earth is estimated as Fe deficient [1]. The solubility and availability of iron in soil can be affected by multiple factors, including soil pH, the redox potential, microbial processes, and the amounts of organic matter and aeration in soil [2]. As a vital cofactor for enzymes, iron takes part in distinct processes, such as facilitating various chemical reactions, modulating protein stability, hormonal regulation, and nitrogen assimilation [1]. Iron deficiency could result in interveinal chlorosis in young leaves as the result of reduced chlorophyll content. The young leaves exhibit yellow color while the veins remain green. All these ultimately lead to the reduction of yield and quality [1,3]. In addition, other nutrients have antagonistic effects on iron uptake, which can significantly reduce the yield of the crops [4]. 

Iron in the rhizosphere is mainly present as Fe^3+^ which is not readily accessible to plants. Different plant species have evolved different strategies for iron acquisition from soil (Figure 1). Non-graminaceous plants, such as tomato and *Arabidopsis*, known as strategy-I plants, use a reduction-based strategy, in which plasma-membrane (PM)-localized H^+^-ATPases (AHAs) release the protons to increase rhizosphere acidification and promote Fe^3+^ solubility. Subsequently, the available ferric Fe^3+^ is reduced to the more soluble ferrous Fe (Fe(II)) by ferric reduction oxidases (FROs) at the apoplast [5]. The reduced ferrous ion (Fe^2+^) is imported into root cells by the Fe^2+^-regulated transporters such as the iron-regulated transporter (IRT1) [6,7]. Additionally, graminaceous plants, including rice, barley, and maize, known as strategy-II plants, use a chelation-based strategy to release phytosiderophores (PS). PS, as strong Fe chelators, are secreted into the rhizosphere with a high affinity for binding Fe (III) [8,9]. PS-Fe(III) is then taken up into root cells through the yellow stripe (YS) or yellow stripe-like (YSL) transporters [10]. 

Iron deficiency triggers the expression of many Fe uptake-associated genes. The expression of *AtAHA2* and *AtAHA7*, for example, are at higher levels under iron-deficient conditions, but *AtAHA1* is not induced by iron deficiency [11]. Twelve PM H^+^-ATPases AHAs are encoded in the *Arabidopsis* genome [11]. AtAHA2 is primarily responsible for the of rhizosphere acidification of root hairs under iron deficiency. Loss function of *AtAHA2* compromised proton extrusion capacity. AHA7 is crucial for the formation of root hairs induced by iron deficiency via mediating H^+^ efflux in the root hair zone. The fine-tuned regulation of root tip H^+^ extrusion by PM H^+^-ATPase is required for root hair formation. H^+^ efflux through PM H^+^-ATPase causes the acidification of the cell wall apoplast, which is crucial for the root hair initiation [11]. The loss function of *AtAHA7* contributed to a decreased frequency of root hairs [11]. However, the mechanism of *AHAs* regulation remains unknown. Recent findings indicate that cytochrome B5 reductase 1 (CBR1) is able to activate plasma membrane-localized H^+^-ATPases, which is achieved by facilitating the content of unsaturated fatty acids [12]. *CBR1* expression is induced under iron-deficient conditions. CBR1 localizes to endoplasmic reticulum (ER) membrane and plays an important role in electron transfer from NADH to cytochrome b5. Then the cytochrome b5 mediates the electrons transfer to fatty acids desaturase 2 (FAD2) and fatty acids desaturase 3 (FAD3), allowing for double bonds into fatty acids. FAD2 is responsible for converting oleic acid (18:1) to linoleic acid (18:2), and FAD3 contributes to the conversion of 18:2 to linolenic acid (18:3). On the other side, 20 or 50 μM of the unsaturated fatty acids 18:2 or 18:3 can strongly activate H^+^-ATPase [12]. Other compounds such as phenolics, organic acids, flavonoids, and flavins have also been implicated in the acidification–reduction strategy to uptake iron (Strategy I) [3,13,14,15]. These small compounds significantly promote reutilization and uptake of apoplastic iron via chelation or the reduction of iron in soil. Recently it was reported that coumarins involved in iron acquisition are secreted and essential for iron uptake under iron-limited conditions [16,17]. The plants are able to secret an array of coumarin-type compounds under different iron nutrition conditions, which facilitate Fe(III) availability [18]. The synthesis of these coumarins require Feruloyl coenzyme A 6’-hydrozylase 1 (F6’H1) enzyme [19]. ATP-BINDING CASSETTE G37 (ABCG37/PDR9) transporters contribute to the exudation of coumarins [17]. Both *F6’H1* and *PDR9* transcript expression are upregulated by iron deficiency [19,20].

Subsequently, the soluble Fe^3+^ is reduced into Fe^2+^ in root apoplast via cellular membrane localized ferric reductase oxidase 2 (FRO2). This protein has 725 amino acids with 8 transmembrane domains, containing motif for binding hemes and NADPH [21]. The electron from NADPH in the cytoplasmic side is transferred via two hemes and Flavin to the Fe^3+^ in apoplast [22]. *FRO2* is primarily expressed in roots [23]. In addition to expression in roots, *FRO2* is largely present in flowers [24]. *FRO2* transcription and post-transcription are both regulated by iron concentration, since the activity of FRO2 in *FRO2* overexpression lines is highly induced under iron deficiency [24]. In addition, iron deficiency facilitates the stability of *FRO2* mRNA [24]. A total of 50 FROs were identified in plants [25] and 8 FROs are encoded in the *Arabidopsis* genome [26]. These FROs have different tissue-specific expression patterns. *AtFRO3* and *AtFRO5* are predominantly expressed in roots, while *AtFRO6*, *AtFRO7* and *AtFRO8* gene expression primarily occur in shoots. *AtFRO1* and *AtFRO4* are present in both roots and leaves [23,27,28,29].

After Fe^3+^ reduced to Fe^2+^ in root rhizosphere, Fe^2+^ can be imported into cells by IRT1 with high affinity to Fe^2+^ (K_m_ = 6 μM). IRT1 is the most important root transporter for ferrous Fe uptake from the soil, while the uptake of other divalent cations (manganese, zinc, cobalt, and cadmium) can also be promoted by IRT1 [6,7,30]. IRT1 is identified in *Arabidopsis* and can rescue the defects of the *fet3fet4* mutants of yeast that are impaired in Fe uptake [6]. The expression of *IRT1* is highly induced under iron-limited conditions [6,7]. IRT1 belongs to ZIP family and consists of 347 amino acids with 8 transmembrane domains. IRT1 can also promote the uptake of and Zn^2+^ but IRT1 can transport Zn only under low pH [30,31]. IRT1 is present in early endosomes/trans-Golgi network compartments (EE/TGN). Early studies found that IRT1 degradation and recycling between EE/TGN and the plasma membrane are modulated by ubiquitination and monoubiquitin-dependent endocytosis [32]. The IRT1 protein can transport to a vacuole for degradation [32]. IRT1 degradation factor1 (IDF1), a RING-type E3 ubiquitin ligase, is found to be responsible for IRT1 ubiquitination on plasma membrane via clathrin-mediated endocytosis. Thus, Fe-deficient induced IDF1 facilitating IRT1 degradation develops a negative feedback loop to fine tune the iron homeostasis [33]. It should be noted that recent studies point to the fact that non-iron elements (Zn, Mn, and Co) are also able to regulate this trafficking of IRT1 between EE/TGN and the plasma membrane in root epidermal cells [34]. Moreover, FYVE1, a phosphatidylinositol-3-phosphate-binding protein, is also required for the recycling of IRT1 and its polar localization to outer polar domain of plasma membrane [34]. SORTING NEXIN (SNX) protein was found to co-localize with IRT1 and is also important for recycling internalized IRT1. In the *snx1* mutants, the degradation of IRT1 is enhanced [35]. Further studies reveal that there exist other transporters for iron uptake. Natural resistance associated macrophage proteins (NRAMPs) were identified as a ubiquitous family of metal efflux transporters. Quite intriguingly, NRAMP1 that acts as a transporter of manganese is also essential for low-affinity iron uptake. Pleckstrin homolog (PH) domain-containing protein AtPH1 binds phosphatidylinositol 3-phosphate (PI3P) in the late endosome, which regulates the localization of NRAMP1 to the vacuole [36].

The strategy II plants, such as rice, can secrete phytosiderophores (PS) in rhizosphere for efficiently increasing the solubility of Fe^3+^, ultimately facilitating the available iron for root acquisition [37]. PS-Fe^3+^ complexes are then imported into root epidermis cells by a specific transporter [37]. PS belong to the family of mugineic acid (MAs), such as mugineic acid (MA), 2’-deoxymugineic acid (DMA), 3-epihydroxymugineic acid (epi-HMA), and 3-epihydroxy 2’-deoxymugineic acid (epi-HDMA) [38,39]. MAs are synthesized from three S-adenosyl-methionine molecules [40]. Yellow stripe 1 (YS1) is firstly identified from maize and targeted to the plasma membrane, which is likely to responsible for transporting Fe^3+^-PS into root cells [10]. YS1 consists of 682 amino acid with 12 transmembrane domains [3]. The transcript expression of *ZmYS1* is highly induced in both root and shoot of maize under iron-deficient condition [10,41]. Eighteen putative yellow stripe 1 (YS1)-like genes (OsYSLs) are identified in the rice genome [42]. 

Fe deficiency readily results in interveinal chlorosis in young leaves, ultimately reducing the yield and grain quality [43]. In order to tolerate iron deficiency, various physiological processes are induced in the root rhizosphere, including ferric reductase activity, the ratio of root and shoot, and photosynthesis. Also, root morphology is altered according to the local availability of iron and for optimizing iron uptake, such as increasing lateral root numbers, extra root hairs, and developing transfer cells to facilitate contact surface with soil [44]. 

## 3. Iron Transport Mechanism in Plants

After iron is transported to the root endodermis from epidermis via apoplastic and symplastic pathway, it needs to be transported to the above ground parts of plants through the xylem (Figure 2). The contents of organic acids, such as citrate, malate, and succinate, are elevated in xylem under iron deficient conditions [45]. The usage of various approaches, such as the theoretical calculations, high-pressure liquid chromatography (HPLC) coupled to electrospray time-of-flight mass spectrometry (HPLC-ESI-TOFMS) and inductively coupled plasma mass spectrometry (HPLC-ICP-MS), detects the natural Fe complex and provides evidence for the transport of iron in xylem to shoots which predominantly occurs as Fe^3+^-citrate complex [46,47,48,49]. The transport of citrate and iron to the xylem is mediated by ferric reductase defective 3 (FRD3) in *Arabidopsis* and its ortholog FRDL1 in rice, which is crucial for iron translocation [50,51]. FRD3 is present only in pericycle and cells neighboring the vascular tissue [50]. *frd3* mutants exhibit severe Fe-deficient phenotype even under Fe-sufficient conditions. Less citrate and less Fe are contained in xylem sap of *frd3* mutants as compared to wild type [50]. *Osfrdl* mutants also contain reduced citrate and Fe in the xylem resembling Fe-deficiency phenotype in *frd3* mutants [52]. Therefore, it is tempted to speculate that graminaceous and nongraminaceous share the similar mechanism by which Fe is transported from root to shoot although the uptake strategies for iron are very different. Ferroportin1 (FPN1) is also responsible for loading iron into the xylem [44]. The *Arabidopsis* genome contains three FPN which have different subcellular localizations. FPN1, for example, is targeted to the plasma membrane, FPN2 on the vacuolar membrane and FPN3 on the chloroplast envelop [44,53,54]. Fe is also capable of translocation in xylem in the form of Fe-nicotianamine (NA) and Fe-MAs. NA as a non-protein amino acid is produced from S-adenosyl methionine by nicotianamine synthase (NAS) and is also the direct biochemical precursor to PS [55,56]. In rice, NA and DMA are present in xylem exudates [57,58]. 

Once the iron reaches the leaves, it must be unloaded to leaf cells from the apoplastic space. NA and DMA are also required for the phloem-based transport [59]. AtYSL1, AtYSL2, and AtYSL3, as metal-NA transporters, are involved in this process, responsible for moving iron from apoplast to symplast [60,61]. These three genes are highly expressed in vascular parenchyma cells of leaves [60,61]. AtYSL2 plays a major role in regulating the lateral distribution of iron from xylem to shoot cells in *Arabidopsis* [54,60]. Moreover, AtYSL1 and AtYSL3 appear to transport the Fe-NA chelate from senescent leaves into the inflorescences and seeds. *ysl1* and *ysl3* mutants contain reduced iron content in leaves and seeds [60,62,63]. In rice, OsYSL2 is likely to be involved in the translocation of Fe(II)-NA to shoots and seeds [42,64]. *OsYSL16* is expressed in the cells surrounding xylem and contributes to Fe(III)-MA allocation via the vascular bundle [65]. OsYSL18 also transports Fe(III)-DMA in reproductive organs and phloem of lamina joints [66]. Recent studies point to OsYSL9 which is involved in the Fe distribution in developing seeds via Fe(II)-NA and Fe(III)-DMA form [67]. Additionally, oligo peptide transporter 3 (OPT3) mediates the Fe transport to sink tissues via the phloem and recirculation in the roots in *Arabidopsis* [68]. Meanwhile, OPT3 is also found to take part in the control of iron movement out of the leaves to root or developing tissues in the form of iron ions rather than iron-ligand complexes [69,70]. Heat shock cognate protein B (HSCB) as a mitochondrial cochaperone participates in iron translocation from roots to shoots [71]. *HSCB* overexpression lines caused iron accumulation in roots but low iron levels in shoots; while *hscb* knockdown plants showed iron accumulation in shoots despite the reduced contents of iron uptake in roots [71]. 

## 4. Iron Storage in Cells

Iron mobilization in cells is essential for plant growth and development, especially under iron-deficient conditions. When transporting across cellular or intracellular membranes, ferric iron is usually reduced to ferrous iron [72]. Iron can produce cytotoxic oxygen radicals, such as hydroxyl radicals and superoxide anions [16]. Generally, the cellular iron is stored in vacuoles and is also likely to be sequestrated into ferritin, which will become available for various metabolic reactions. In *Arabidopsis* seeds, the vacuole is the major iron store containing about 50% of total iron, while ferritins play a minor role in iron storage including about 5% iron [16,73]. Ferritin is important for fine tuning the quantity of metal which is required for metabolic purposes [74]. In the vacuole of *Arabidopsis* seeds, globoids act as an important site for Fe storage [16]. However, in pea, the amount of iron-ferritin is present at about 92% of the total seed iron in embryo axis [75]. Therefore, these findings suggest that the way for iron storage in seeds may be different between different species, such as *pea* and *Arabidopsis* [73]. Plastids also act as a sink for iron in cells and appear to function in sensing and maintaining iron concentration in the plants to adapt various changes [76]. In chloroplast, ferritins represent one candidate to form the complex with Fe [76]. In *Arabidopsis*, three of ferritins are localized to chloroplasts. In addition, NA might also play a role in maintaining Fe soluble in plastids [76].

The changes of iron content in vacuole might trigger distinct responses. The vacuolar iron transporter 1 (VIT1), an orthologue of the yeast iron transporter Ca^2+^-sensitive cross-complementer 1 (CCC1), was first identified in *Arabidopsis* [77]. AtVIT1 was found to control iron sequestration into vacuoles. Despite there being no difference in the iron content of seeds between *vit1* mutants and wild type, the iron accumulation is absent in the vacuoles of provascular cells [77]. So, what else could modulate iron mobilization efflux from vacuolar? AtNRAMP3 and AtNRAMP4 are responsible for Fe efflux from the vacuolar into the cytosol, and consequently essential for seed germination under Fe deficiency [78,79]. However, we cannot exclude other efflux transporters localized in vacuolar. In rice, the molecular mechanism underlying Fe transport in cells has also been well uncovered. OsVIT1, OsVIT2, and OsNRAMPs affect Fe translocation from the vacuole to other parts [80,81,82,83]. 

Ferritins, as another iron pool, are a class of universal 24-mer multi-meric, which are encoded by nuclear genes [84]. The structure of ferritins is highly conserved in eukaryotes [74]. In *Arabidopsis*, four ferritin genes (AtFer1–4) have been identified, among which FER1, FER3, and FER4 are proposed to exist in leaves while FER2 is present in seeds [74]. Recent studies found that ferritins are vital for protecting cells against oxidative stress [73]. Recently it was reported that ferritins are also involved in root system architecture regulation. Triple mutants of *fer1 fer3 fer4* exhibited altered root architecture which was caused by the alteration in the production and balance of reactive oxygen species (ROS) [85]. 

In addition, mitochondrion as a crucial iron sink provides available iron for the proper respiration. In rice, FRO3 and FRO8 appear to play roles in Fe*^3+^* reduction in the mitochondrial membrane and mitochondrial iron transporters (MITs) are responsible for the translocation of iron from cytoplasm to mitochondrial [86]. Although the total iron content of shoots is increased in *mit* knockdown mutants as compared to wild type, the iron concentration in mitochondria is reduced, which further suggest iron is mistransported in the mitochondria of these mutants. Additionally, *mit* knockdown mutants contain a significant reduction of chlorophyll content and impair plant growth [87]. 

Also, chloroplast represents one of the main sinks for iron in plant cells. The iron transport across the chloroplast inner envelope also depends on reduction-based strategy. AtFRO7 as a chloroplast Fe (III) chelate reductase is targeted to the chloroplast envelope and putatively function in Fe^3+^ reduction in chloroplast. AtFRO7 is required for the survival of young seedlings under iron-deficient conditions. Under Fe-deficient conditions, loss of function of *FRO7* reduces the Fe content and hampers the reductase activity of chloroplast, leading to chlorotic appearance [29]. AtYSL6 is localized to the chloroplast envelope. Plants lacking *ATYSL4* and *ATYSL6* exhibit iron over-accumulated chloroplasts and the overexpression lines are characterized by decreased Fe content in chloroplast, suggesting that YSL4 and YSL6 take part in the release of iron from chloroplast [88]. In addition, PERMEASE IN CHLOROPLASTS1 (PIC1) as an ancient permease plays a role in chloroplast Fe uptake and maintaining Fe homeostasis. Interestingly, PIC1 was identified as the first protein involved in Fe uptake in plastid [89], which is localized to the inner envelope and contain four membrane-spanning α-helices [89]. The *pic1* mutant exhibits altered mesophyll organization, disrupted chloroplast and thylakoid development, which is consistent with Fe-deficiency phenotype [89]. Furthermore, recent findings further confirm this function of PIC1 in plastid Fe-transport using *PIC1* knockdown and overexpression lines in *Nicotiana tabacum* [90]. 

## 5. Transcriptional and Posttranscriptional Regulation of Fe-related Genes

Since Fe is vital for cellular process, a sophisticated regulatory mechanism to sense and adjust iron deficiency is essential for providing sufficient iron for plant growth and development. To avoid iron deficiency, various genes involved in iron acquisition and internal translocation are fine-tune regulated at the transcriptional and posttranscriptional level in adapt to iron deficient condition (Figure 1). Fe efficiency reactions (FER) was firstly identified in tomato and encoding a bHLH transcription factor. In this regard, FER controls the root physiology and morphology adapt to iron deficiency [91]. The basic helix-loop-helix (bHLH) FER-like iron deficiency-induced transcription factor (FIT) was identified in *Arabidopsis* and involved in iron sensing, responding, and acquisition through regulating the expression of FRO2 and IRT1 [92]. The ethylene-responsive transcription factors Ethylene Insensitive3 (EIN3) and EIN3-Like1 (EIL1) both enable interact with FIT, consequently activating FIT [93]. The activated FIT can up-regulate the transcript expression of AHA2, FRO2, and IRT1 [94,95,96]. Extensively, FIT activity is modulated via interaction with other proteins. The expressions of bHLH038, bHLH039, bHLH100, and bHLH101 have been reported to be increased under Fe starvation and interact with FIT [97,98]. These interactions result in the activation of FIT and consequently activate the expression of FIT target genes such as IRT1 and FRO2 [98,99]. However, the transcript expression of NRAMP3 is not influenced by the activated FIT [94]. What is more, the activity of FIT can be inhibited by the interaction of DELLA with FIT [100]. In addition, a bHLH transcription factor POPEYE (PYE) is identified which as part of an iron regulatory network is independent of FIT. PYE is capable of interacting with another PYE homologs-bHLH transcription factor IAA-Leu Resistant3 (ILR3), which regulates the iron deficiency response and are both required for maintaining iron homeostasis [101]. Under low iron conditions, PYE is expressed in the root vasculature, columella root cap, and also lateral root cap. Interestingly, its strongest expression occurs in the pericycle of the maturation zone [101]. A putative E3 ligase protein BRUTUS (BTS) can also interact with ILR3, but plays a negative role in response to iron deficiency [101]. Also, the transcription factors, MYB family members MYB10 and MYB72, are implicated in the regulation of NAS4 expression [102,103]. WRK46 not only regulates the expression of NAS but also enables to directly bind the promoter of VIT-LIKE1 via the W-boxes, thereby controlling the iron translocation [104]. YSL2 expression can be controlled by the transcription factors IDEF1 and IDEF2 in rice [105]. In rice, Fe-deficiency-inducible bHLH transcription factor OsIRO2, as the homologue of AtbHLH39, enhances the expression of YSL15 [106]. 

## 6. Function of Other Factors in the Iron Homeostasis

Although recent studies have demonstrated Fe-related genes are associated with plants response to Fe deficiency, the reality of signal network appears to be more complicated in adapting to iron deficiency. Various plant hormones, messenger molecules and kinases are implicated into this process. Auxin analogs for example can increase the activity of the root ferric chelate reductase (FCR) in bean [107,108,109]. In *Arabidopsis*, abscisic acid (ABA) and gibberellin have been suggested to facilitate the Fe deficient response, while cytokinin and jasmonic acid prevent this response [110,111,112,113]. ABA, for example, promotes the secretion of phenolics and also iron efflux from vacuole via up-regulation of AtNRAMP3. Further studies suggest that ABA enhances the Fe translocation from root to shoot [110]. Nitric oxide (NO) is also be found to act as a component of Fe signal pathway and activate root FCR activity under iron deficiency via acting downstream of auxin in *Arabidopsis* [114]. NO plays a role in the synthesis of cell wall. Cell wall consists of pectin, cellulose, and hemicellulose. Cell walls are full of negative charges, which provide the binding sites for metal ions. Pectin is secreted into the apoplast from the symplast. Pectin methylesterase (PME) contributes to de-methylation of pectin that can increase carboxylic groups and hence provides more negative charged sites for iron in cell wall. Fe-deficiency induced NO prevents pectin methylation of cell wall and stimulates the PME activity. These together enhance the Fe retention in root apoplast. In this regard, NO limits iron translocation from root to shoot [115]. Recent evidence points to Ca^2+^ direct interrelations of Fe signal. An important signaling network in deciphering Ca^2+^ signals is formed by specific interactions of 10 calcium B-like proteins (CBLs) and 26 CBL interacting protein kinases (CIPKs) in *Arabidopsis* [116,117]. CIPK23 could be as “nutritional sensors” to sense and mediate the iron homeostasis in *Arabidopsis*. *cipk23* mutants exhibit lower activity of FCR and the regulation of FCR activity by CIPK23 is not related to the transcript expression of *FRO2*, *FRO3,* and *FRO5* [118]. Additionally, it has been found that CIPKs are also involved in the regulation of H^+^ homeostasis. CIPK11/PKS5 suppresses the activity of the PM H^+-^ATPase (AHA2) via phosphorylation which prevents the interaction between AHA2 and 14-3-3 protein, and thus inhibits the extrusion of protons (H^+^) to the extracellular space [119]. Moreover, CIPK11 interacts with FIT and activates FIT via phosphorylation at Ser272, allowing for FIT-dependent Fe deficiency responses. Mutation at Ser272 of FIT affects seed iron content [120]. 

## 7. Conclusions

Iron acts as an essential element not only in plant physiological functions but also in the maintenance of various cell processes. Over the past decades, accumulating progresses have been achieved in understanding how the plants adapt to iron deficiency in soil. Cellular, biochemical, molecular, genetics, and genomic approaches facilitate a better understanding of iron uptake, transport, and utilization. However, how to observe the iron dynamics in plants, especially in different tissues and cells, is still a notable challenge. Despite a wealth of information pointing to the identities for many genes responsible for iron uptake from soil, transport from roots to shoots, storage in cells, and even their regulation at the transcription and post-transcription level, further research is clearly needed to uncover the further interconnection and integration of signaling pathways of iron deficiency into development and physiological networks. Finally, all of this information underlying the mechanism of iron uptake, transport, and homeostasis will be of great benefit to plants and human health. 

## Figures and Tables

**Figure 1 ijms-20-02424-f001:**
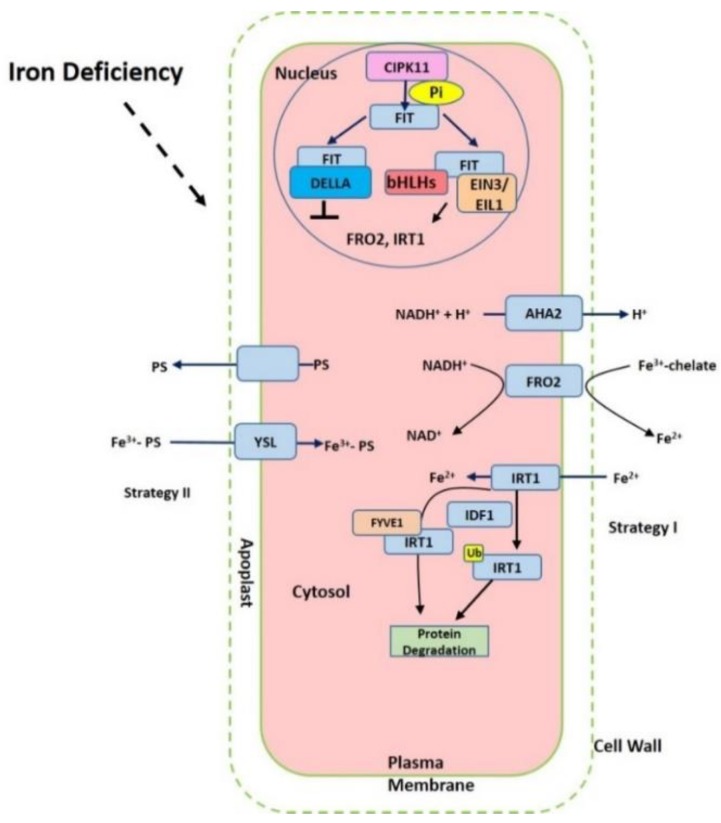
Summary of the iron-deficient response in plant cells. The proton ATPase AHA2, Ferric chelate reductase FRO2 (ferric reduction oxidase), Fe^2+^-regulated transporters iron-regulated transporter (IRT1) and FER-like iron deficiency-induced transcription factor (FIT) are activated under iron starvation, respectively. AHA2 (H^+^-ATPase) increases the acidification of rhizosphere to facilitate iron solubilization. FRO2 reduces ferric iron to ferrous iron that is imported into the cell via IRT1. The expression of *FRO2*, *IRT1* can be induced via FIT interaction with other transcription factors such as bHLHs and EIN3/EIL1 but prevented with DELLA.

**Figure 2 ijms-20-02424-f002:**
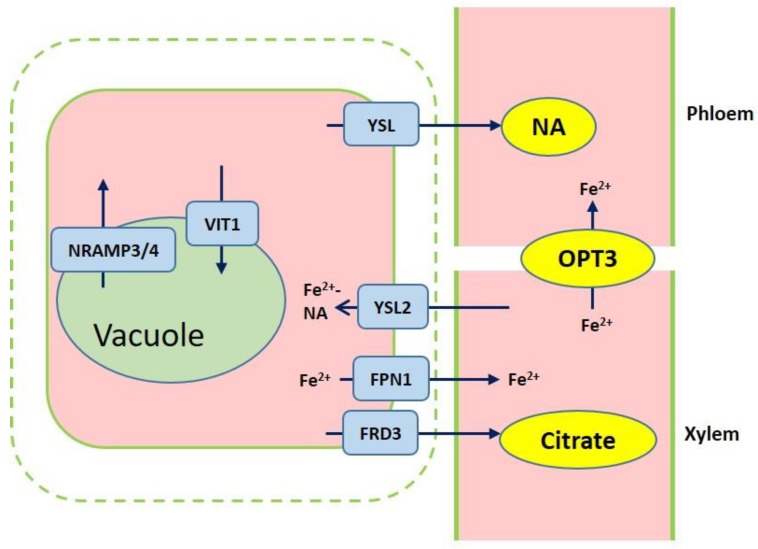
Overview of iron transport from roots to shoots. Ferric reductase defective 3 (FRD3) and ferroportin1 (FPN1) are responsible for importing citrate and iron into the xylem. Iron chelation with citrate or NA are translocated to shoots. Yellow stripe-like 2 (YSL2) contributes to the Fe^2+^-NA distribution from the xylem to neighboring cells. Iron is loaded into vacuole through VIT1, while iron efflux of vacuolar occurs via NRAMP3 and NRAMP4. OPT3 mediates the Fe transport to sink tissues via the phloem.

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
