# Peer review of "The Adaptive Mechanism of Plants to Iron Deficiency via Iron Uptake, Transport, and Homeostasis"

_ijms, 2019, doi:10.3390/ijms20102424_

Reviewer 1 Report

In their work entitled "The adaptive mechanism of plants to iron deficiency ", Xinxin Zhang and colleagues review the transport and assimilation by irons.

The review is well written and the reader will find relevant information related on iron transport and homeostasis in the plant.

My only remark would be that the title is a little bit misleading, since the authors discuss on plant adaptation to iron deficiency at the end only, and in my opinion the review is more on iron homeostasis.

Author Response

Comments and Suggestions for Authors  In their work entitled "The adaptive mechanism of plants to iron deficiency ", Xinxin Zhang and colleagues review the transport and assimilation by irons.

The review is well written and the reader will find relevant information related on iron transport and homeostasis in the plant.

We thank this reviewer for the constructive comments and thank you very much for the positive feedback.

My only remark would be that the title is a little bit misleading, since the authors discuss on plant adaptation to iron deficiency at the end only, and in my opinion the review is more on iron homeostasis.

We apologize that our favored explanation was not formulated clearly enough. We therefore edited the tittle to “The adaptive mechanism of plants to iron deficiency via iron uptake, transport and homeostasis”. In this way, the content is better to fit the title.

Reviewer 2 Report

The review entitled ‘The adaptive mechanism of plants to iron deficiency’ is well written. This reviewer however recommends the authors to address several concerns as described below for the publication in International Journal of Molecular Sciences.

(1) The title of ‘Section 2’ is missing between ‘1. Introduction’ and ‘3. Iron transport mechanism in plants’.

(2) l. 77-79 

What is the mechanism by which AtAHA7 contributed to the formation of root hairs ?

The readers would like to know the thought of the authors of this literature.

(3) l. 79-82

There is no description of how CBR1 leads to a quantitative increase of unsaturated fatty acids, and then to the activation of plasma membrane-localized H+-ATPase. This explanation is necessary for the readers to understand the important role of CBR1.

(4) l. 105

What is ‘Km-10mM’?

(5) l. 121

What is ‘outer plasma membrane’?

(6) l. 200-209

Free Fe ion is toxic due to its probable involvement in Fenton reaction that produces ROS. The formation of the complex of Ferritin/Fe would be one of strategies to avoid this ROS-generating reaction. This reviewer recommends the authors to explain how vacuoles or chloroplasts stock Fe to repress ROS generation.

(7) l. 294-296

It is difficult for the readers to understand the role of NO signaling in Fe translocation. What is the role of NO in the cell wall synthesis? Why does prevention of the pectin methylation of cell wall lead to iron retention in root apoplasts?

Author Response

The review entitled ‘The adaptive mechanism of plants to iron deficiency’ is well written. This reviewer however recommends the authors to address several concerns as described below for the publication in International Journal of Molecular Sciences.

Thank you very much for this positive feedback.

The title of ‘Section 2’ is missing between ‘1. Introduction’ and ‘3. Iron transport mechanism in plants’.

The title of “Section 2” is in the line of 45.

(2) l. 77-79 

What is the mechanism by which AtAHA7 contributed to the formation of root hairs?

The readers would like to know the thought of the authors of this literature.

Thank you for this comment. We have now added this part of our explanation to follow this advice (Lines 80-83). We also change the literature of 11 to the recently published literature (Lines 369-372).

(3) l. 79-82

There is no description of how CBR1 leads to a quantitative increase of unsaturated fatty acids, and then to the activation of plasma membrane-localized H+-ATPase. This explanation is necessary for the readers to understand the important role of CBR1.

As requested, we now include the detailed explanation of the CBR1 role in unsaturated fatty acids and the relationship of unsaturated fatty acids and H+-ATPase activity (Lines 87-93).

(4) l. 105

What is ‘Km-10mM’?

We apologize that the Km is not clearly explained. The form and the number is written wrong. It should be Km= 6 μM. It means the high affinity of IRT1 to iron (Line 117).

(5) l. 121

What is ‘outer plasma membrane’?

Thank you for this question.“outer plasma membrane” has been changed to “outer polar domain of plasma membrane” (Line 133).

(6) l. 200-209

Free Fe ion is toxic due to its probable involvement in Fenton reaction that produces ROS. The formation of the complex of Ferritin/Fe would be one of strategies to avoid this ROS-generating reaction. This reviewer recommends the authors to explain how vacuoles or chloroplasts stock Fe to repress ROS generation.

According to this comment we have now extend the explanation in Lines 213-214, Line 218-219 and Lines 223-226.

(7) l. 294-296

It is difficult for the readers to understand the role of NO signaling in Fe translocation. What is the role of NO in the cell wall synthesis? Why does prevention of the pectin methylation of cell wall lead to iron retention in root apoplasts? 

We apology that our explanation is not clearly. More explanations have been added in Lines 314-320 as suggested by the reviewer.

This manuscript is a resubmission of an earlier submission. The following is a list of the peer review reports and author responses from that submission.